# Two pathways are required for ultrasound-evoked behavioral changes in *Caenorhabditis elegans*

Uri Magaram[1,2], Connor Weiss[2], Aditya Vasan[3], Kirthi C. Reddy[2], James Friend[3], Sreekanth H. Chalasani[2]*

1 Neurosciences Graduate Program, University of California San Diego, La Jolla, CA, United States of America, 2 Molecular Neurobiology Laboratory, The Salk Institute for Biological Studies, La Jolla, CA, United States of America, 3 Medically Advanced Devices Laboratory, Department of Mechanical and Aerospace Engineering, Jacobs School of Engineering and Department of Surgery, School of Medicine, University of California San Diego, La Jolla, CA, United States of America

* schalasani@salk.edu

**Data Availability Statement:** All relevant data are within the paper and its Supporting Information files.

## Abstract

Ultrasound has been shown to affect the function of both neurons and non-neuronal cells, but, the underlying molecular machinery has been poorly understood. Here, we show that at least two mechanosensitive proteins act together to generate *C. elegans* behavioral responses to ultrasound stimuli. We first show that these animals generate reversals in response to a single 10 msec pulse from a 2.25 MHz ultrasound transducer. Next, we show that the pore-forming subunit of the mechanosensitive channel TRP-4, and a DEG/ENaC/ASIC ion channel MEC-4, are both required for this ultrasound-evoked reversal response. Further, the *trp-4;mec-4* double mutant shows a stronger behavioral deficit compared to either single mutant. Finally, overexpressing TRP-4 in specific chemosensory neurons can rescue the ultrasound-triggered behavioral deficit in the *mec-4* null mutant, suggesting that both TRP-4 and MEC-4 act together in affecting behavior. Together, we demonstrate that multiple mechanosensitive proteins likely cooperate to transform ultrasound stimuli into behavioral changes.

## Introduction

Ultrasound has been shown to modify neuronal activity in number of animal models including humans [1, 2]. However, the direction of this action is somewhat controversial with some reporting activation [3–7], while others demonstrate inhibition [2, 8–10]. Moreover, the underlying mechanisms for ultrasound action on neuronal membranes have been suggested to include thermal [11–13], mechanical (direct or via cavitation [14–16]) or a combination of two [17]. Additionally, ultrasound neuromodulation has also been shown to include astrocyte signals *in vitro* [18] and auditory signals *in vivo* [19, 20]. To identify the underlying molecular mechanisms, we and others have been examining how ultrasound affects neurons in tractable invertebrate systems [16, 21–23] or mammalian cell [24, 25] and slice cultures [3, 26].

**Funding:** S.H.C. R01MH111534, R01NS115591 National Institutes of Health www.nimh.nih.gov, www.ninds.nih.gov The funders had no role in study design, data collection and analysis, decision to publish, or preparation of the manuscript.

**Competing interests:** The authors have declared that no competing interests exist.

The nematode, *C. elegans* with just 302 neurons connected by identified chemical and electrical synapses generating robust behaviors with powerful genetic tools is ideally suited to probe the molecular effects of ultrasound on neuronal membranes [27–29]. We previously showed that the pore-forming subunit of the mechanosensitive ion channel TRP-4 is required to generate behavioral responses mediated by microbubbles activated by a single 10 ms pulse of ultrasound generated from a 2.25 MHz focused transducer [21]. This channel is specifically expressed in few dopaminergic (CEPs, and ADE) and interneurons (DVA and PVC) in *C. elegans*, where it has been shown to be involved in regulating the head movement and locomotion [30, 31]. Surprisingly, we found that ectopically expressing this TRP-4 in a neuron rendered that neuron sensitive to ultrasound stimuli, confirming ultrasound-triggered, microbubble-mediated control [21]. Moreover, a second mechanosensitive protein, a DEG/ENaC/ASIC ion channel MEC-4 was also shown to be required for behavioral responses to a 300 ms duration, 1 KHz pulse repetition frequency at 50% duty cycle generated from a 10 MHz focused ultrasound transducer [22]. These two studies confirm that ultrasound effects on *C. elegans* behavior is likely mediated by mechanosensitive proteins.

In this study, we used genetic tools in *C. elegans* to test whether TRP-4 and MEC-4 are both required to mediate the behavioral effects of ultrasound stimuli. We generated a *trp-4 mec-4* double mutant and compared its ultrasound responses to both single mutants. Also, we found that ectopically expressing TRP-4 in specific chemosensory neurons can rescue the behavioral deficits in both *trp-4* and *mec-4* null mutants, confirming that these genes act together to drive ultrasound-evoked behavior. Our study demonstrates that multiple mechanosensory pathways act in concert to generate behavioral responses to ultrasound stimuli.

## Results

To test the behavioral responses of *C. elegans* to various ultrasound stimuli, we aligned a transducer with a holder that positioned agar plates at the water level in a tank (**Fig 1A**). Animal responses were captured using a camera and analyzed (**Fig 1B and 1E**, See Methods for more details). Next, we evaluated both pressure and temperature changes at the agar surface for a single 10 ms pulse of ultrasound stimuli of different intensities. We assessed the area on the agar surface which was ensonified by the ultrasound stimuli and found that our system delivered mechanical, but not temperature changes (**Fig 1C and 1D**). We found that ultrasound stimuli delivering peak negative pressures greater than 0.75 MPa amplified by 1–10 μm-sized gas filled microbubbles generated robust responses in wild-type (WT) animals. We analyzed these responses and found that animals generated robust increases in their large reversals (events where the head bends twice or more), but not omega bends or small reversals (where the head bends only once) (**Figs 2 and S1 and S1 Video**). These data are consistent with previous studies showing that *C. elegans* generates dose-dependent responses to ultrasound stimuli [21, 22].

We then analyzed the behavioral responses of null mutants in both the TRP-4 and MEC-4 channels to ultrasound. We found that *trp-4(ok1605)* and *mec-4(u253)* mutants are both defective in their ultrasound-evoked large reversal behavior both with and without gas-filled microbubbles (**Fig 2A and 2B**). Also, we observe that *mec-4(u253)* null mutants have significant defects at higher ($>0.92$ MPa), but not lower pressures generated from a 2.25 MHz ultrasound transducer (**Fig 2A and 2B**). Similarly, we also found that an outcrossed allele of *trp-4(ok1605 4X)* was also defective in its response to ultrasound stimuli at higher pressures (**Fig 2A and 2B**). This result is consistent with our previous study, which identified a critical role for *trp-4* in mediating ultrasound-evoked behavioral responses [21]. Additionally, we found that *C. elegans* can also respond to ultrasound stimuli even in the absence of gas-filled microbubbles, consistent with a previous study [22]. Moreover, we find that at least at lower ultrasound

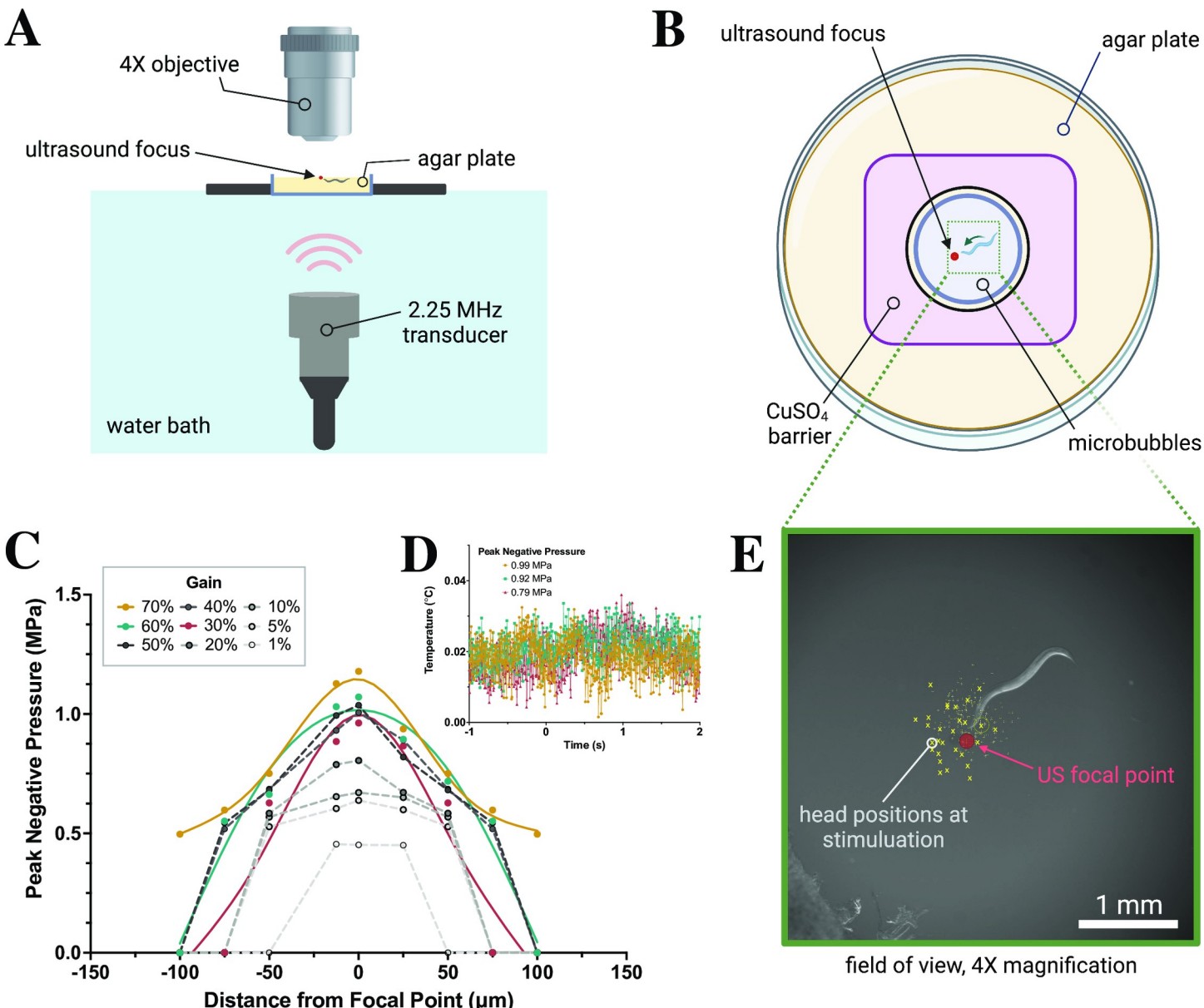

**Fig 1. Recording *C. elegans* behavior in response to ultrasound at 2.25 MHz.** (A) Schematic of 2.25 MHz ultrasound imaging system with transducer, water bath, and 4x objective over agar plate. (B) Top view of agar plate with animals corralled by copper sulfate barrier (1.5 cm in diameter) on agar plate with polydisperse microbubbles. (C) Fiberoptic hydrophone measurements at perpendicular distance from focal point of transducer show peak negative pressures ~1 MPa, with (D) negligible temperature changes at 10ms ultrasound pulses at t = 0. Individual points connected via spline fit. (E) Relative head positions (yellow dots) of each animal at time of stimulation, some example points highlighted for visibility.

pressures (0.79 MPa), the probability of *C. elegans'* responses are increased in the presence of microbubbles (**Fig 2A and 2B**). Importantly, we found that at high pressure, the *trp-4(ok1605 4X); mec-4(u253)* double mutant had a stronger defect in ultrasound-trigged large reversal responses compared to either single mutant (**Fig 2C and 2D**), both with and without micro-bubbles (**Fig 2C and 2D**), suggesting that these genes likely act together. Moreover, we do not observe any consistent change in small reversals, however changes in omega bends were simi-lar to what we found with large reversals (**S1 Fig**). These data are consistent with previous

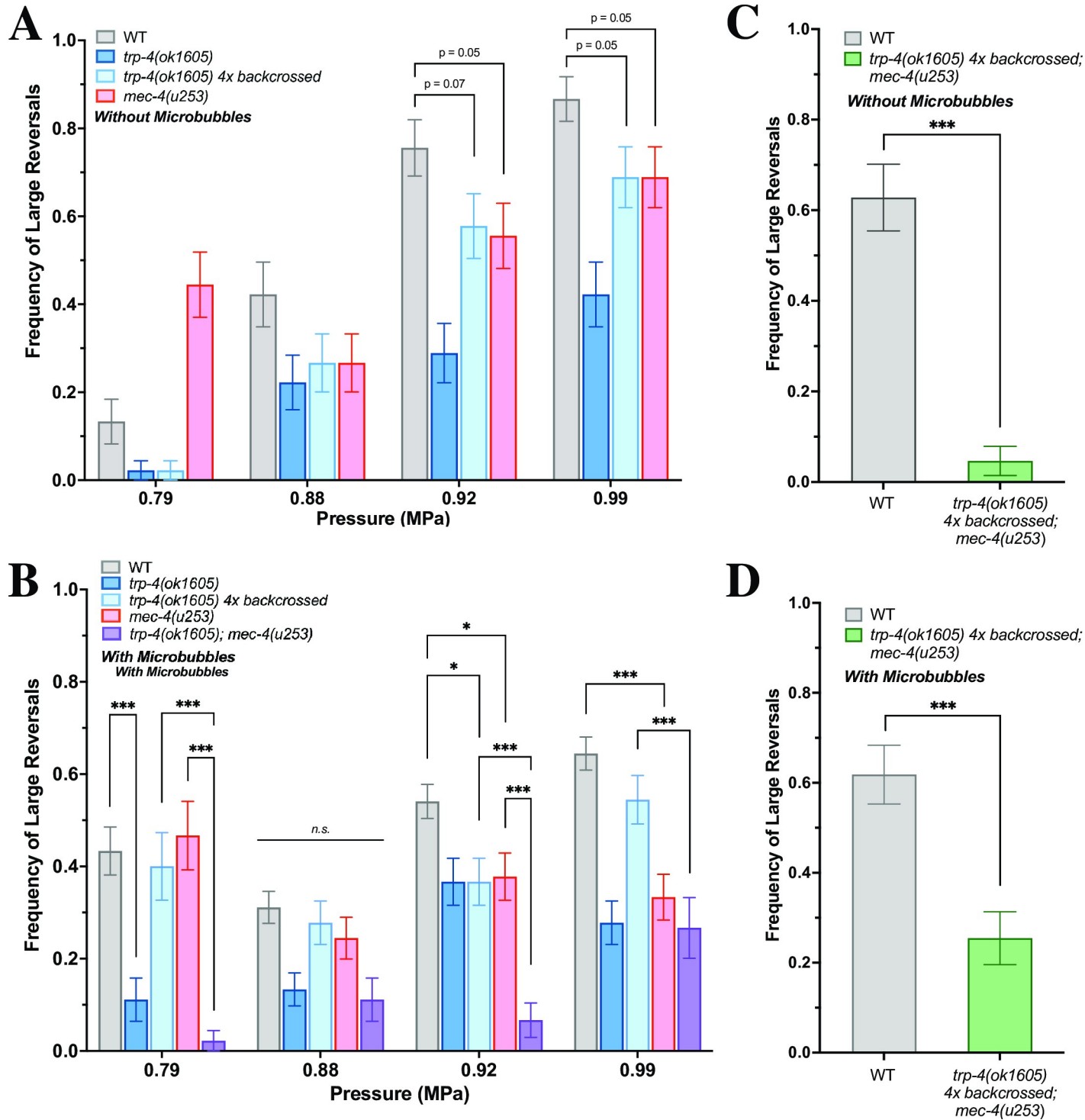

**Fig 2. Mutants in mechanosensitive proteins are defective in their responses to ultrasound. Frequency of large reversals (more than two head bends) with and without microbubbles.** Large reversal frequency (A) without microbubbles and (B) with microbubbles at various peak negative pressures are quantified. n = 90–135 for each condition, n = 45 for double mutant strains. At certain pressures, the double mutant *trp-4;mec4* responds significantly less than each of the individual mutants, which respond less than WT animals. At 0.99 MPa, the outcrossed double mutant *trp-4 4x;mec-4* shows a significant defect in large reversals both without (C, n = 43) and with (D, n = 55) microbubbles. Proportion of animals responding with standard error of the proportion are shown. ***$p < .001$, **$p < .01$, *$p < .05$ by two-proportion z-test with Bonferroni correction for multiple comparisons, with c = 2 for 2A, c = 5 for 2B.

studies showing that omega bends often occur together with large reversals [32, 33]. Collectively, these data indicate that MEC-4 and TRP-*4* channel proteins might be acting together to mediate *C. elegans* behavioral responses to ultrasound stimuli.

To confirm whether these two mechanosensitive proteins are acting together, we tested combinations of transgenic animals ectopically expressing TRP-4 in various null mutant backgrounds. We expressed TRP-4 under ASH and AWC-chemosensory neuron selective promoters and analyzed the ultrasound responses of the resulting transgenics [21]. Neither ASH nor AWC expression of TRP-4 in wildtype animals altered its ultrasound-behavior (**Fig 3A and 3B**). However, at lower pressures (0.79 MPa) TRP-4 expression in AWC chemosensory neurons was able to partially rescue large reversals in both *trp-4(ok1605)* and *mec-4(u253)* mutants (**Fig 3A**). In contrast, we found that ASH expression of TRP-4 was able to partially rescue behavioral deficits in both the *trp-4(ok1605)* and *mec-4(u253)* null mutants only at higher pressures (> 0.92 MPa) (**Fig 3B**). These data suggest that TRP-4 protein likely functions in different neurons to affect ultrasound-evoked large reversals: AWC for lower pressures and ASH at higher pressures (**Fig 3A and 3B**). While AWC neurons are known to have an expanded fan-shaped cilia, ASH neurons have a rod shaped cilia [34]. We suggest that this difference in the shape of the cilia might result in AWC and ASH neurons having a different sensitivity to ultrasound-evoked stimuli. In addition, we found that AWC and ASH-selective expression of *trp-4* was able to rescue the behavioral deficits observed in the *trp-4(ok1605 4x)* outcrossed strain (**Fig 3C**) confirming that TRP-4 likely functions in these two chemosensory neurons to drive ultrasound-evoked large reversal behaviors. Also, while small reversals were not consistent, omega bends often matched large reversals (**S2 Fig**). Collectively, these data suggest that TRP-4 and MEC-4 likely act together to generate large reversals in response to ultrasound stimuli.

## Discussion

We showed that a double mutant that deleted both MEC-4 and TRP-4 channels had stronger behavioral deficits compared to either single mutant alone. Additionally, we showed that AWC and ASH-specific expression of TRP-4 could partially rescue the deficit in both *mec-4 (u253)* and *trp-4(ok1604 4x)* single mutants confirming that these two pathways act together to drive ultrasound-evoked changes in large reversals. We suggest that expression of TRP-4 channel in AWC and ASH neurons renders them sensitive to ultrasound stimuli. Activating these neurons has been previously shown to generate reversal behavior [35, 36]. We speculate that the ultrasound-driven activation of ASH and AWC neurons mediated by TRP-4 channels acts independent of the MEC-4 pathway to generate reversal behavior.

The nematode *C elegans* has provided insights into our understanding of how ultrasound affects animal behavior. We previously showed that ultrasound evoked behavioral changes required the pore-forming subunit of the TRP-4 mechanosensitive channel [21]. This protein is selectively expressed in a few dopaminergic and interneurons and is likely involved in generating head movement and coordinating locomotory behaviors. We suggested that delivering ultrasound to the head of the animal likely activates this channel resulting in reversal behavior [21]. Moreover, a second mechanosensitive channel, MEC-4 (DEG/ENaC/ASIC) has been shown to be required for ultrasound-evoked behavioral responses in *C. elegans* [22]. MEC-4 is a key component of the touch sensitive mechanosensitive ion channel and is expressed in the ALM, PLM, AVM, PVM, FLP and other touch-activated neurons [37, 38]. This study indicated that ultrasound delivered to the head of the animal would also generate a reversal response [22]. While we find that *mec-4(u253)* and *trp-4(ok1605 4X)* mutants are indeed defective under our stimulus conditions and likely act together to generate ultrasound-evoked behavioral changes, we are unable to observe ultrasound-evoked behavioral changes to 10 MHz

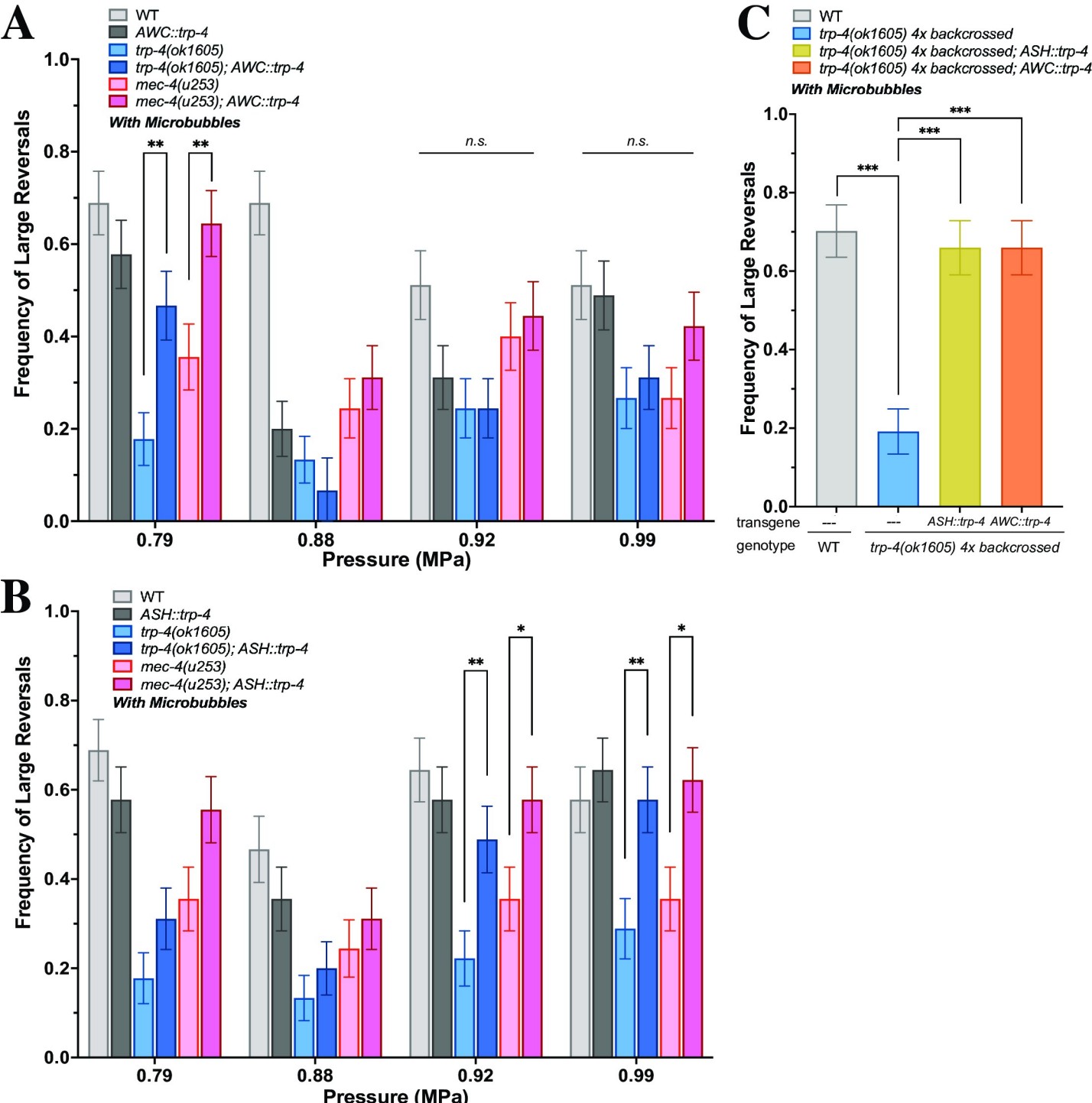

**Fig 3. MEC-4 and TRP-4 act in parallel to mediate ultrasound-evoked behavioral changes.** Frequency of large reversals (more than two head bends) in strains ectopically expressing TRP-4. Large reversal frequency with (A) *AWC::trp-4* and (B) *ASH::trp-4* in different backgrounds at different peak negative pressures are quantified. At certain conditions, trp-4 expression significantly increases large reversal frequency in both *trp-4* knockout and *mec-4* knockout animals. (C) At 0.99 MPa, TRP-4 expression in both ASH and AWC rescues large reversal behavior in the outcrossed *trp-4(ok1605)4x* mutant. n = 45 for each condition. Proportion of animals responding with standard error of the proportion are shown. *** $p < .001$, ** $p < .01$, * $p < .05$ by two-proportion z-test.

ultrasound stimuli (S3 Fig). Although we were unable to achieve the magnitude of behavior reported in [22], we observed that, consistent with our other findings, introducing microbubbles to the assay did significantly increase worm responses to ultrasound stimuli (S3C Fig). Perhaps, differences in ultrasound delivery and stimulus parameters might explain the discrepancy between our study and previous studies, which have reported *C. elegans* responses to 10 MHz and ~28 MHz [22, 39, 40]. Additionally, while previous studies have shown that *mec-4 (u253)* are defective to a broad range of ultrasound pressures delivered from 10 MHz transducer [22], we observe that these *mec-4(u253)* mutants are only defective at pressures greater than 0.92 MPa delivered from a 2.25 MHz transducer. These data might imply that distinct molecular machinery might sense ultrasound stimuli at different frequencies. Furthermore, we find that *trp-4(ok1605 4x)* has similar behavioral deficits compared to *mec-4(u253)* animals confirming that these two genes might act together to affect ultrasound-evoked behaviors.

Ultrasound has been used to non-invasively manipulate both neuronal and non-neuronal cells in a number of animals including humans. We show that animals missing two mechanosensitive proteins are defective in their responses to ultrasound stimuli, confirming a role for mechanosensation in mediating the biological effects of this modality. This is also consistent with multiple studies identifying other mechanosensitive proteins that can confer ultrasound sensitivity to mammalian cells *in vitro* and *in vivo* [24–26, 41–43]. This is particularly relevant to the method of using ultrasound to selectively and non-invasively manipulate cells within an animal ("Sonogenetics"). Furthermore, our study implies that ultrasound might affect at least two mechanosensitive ion channels to affect animal behavior. Identifying downstream signaling pathways of these and other ultrasound-sensitive mechanosensitive channels would provide a framework to decode ultrasound neuromodulation and enhance sonogenetic control.

## Methods

### Ultrasound imaging assay

A schematic of the system for imaging ultrasound-triggered behavior appears in Fig 1A. An immersible 2.25 MHz central frequency point focused transducer (V305-SU-F1.00IN-PTF, Olympus NDT, Waltham, MA) was positioned in a water bath below a 60 mm 2% agar-filled petri dish, and connected via waterproof connector cable (BCU-58-6W, Olympus). A single 10ms pulse was generated using a TTL pulse to trigger a multi-channel function generator (MFG-2230M, GWINSTEK, New Taipei City, Taiwan); the amplitude of the signal was adjusted through a 300-W amplifier (VTC2057574, Vox Technologies, Richardson, TX) to achieve desired pressures. *C. elegans* behavior was captured through a high speed sCMOS camera (Prime BSI, Photometrics, Tuscon, AZ) and a 4x objective (MRH00041, Nikon, Chicago, IL). Worm movement was compensated via a custom joystick-movable cantilever stage (LVP, San Diego, CA) controlled by Prior Box (ProScanIII, Prior, Cambridge, UK). All components were integrated via custom MetaMorph software (Molecular Devices, San Jose, CA). A goose-neck lamp (LED-8WD, AmScope, Irvine, CA) at ~45° provided oblique white light illumination.

The ultrasound transducer was focused in the Z-plane to the focal plane of the camera, allowing for X-Y motion of the stage and petri dish using the joystick-movable stage all in the focal plane of both the ultrasound transducer and camera. The petri dish was coupled to the ultrasound transducer via degassed water in the water bath.

For 10 MHz experiments, the 2.25 MHz transducer was replaced with a 10 MHz line-focused transducer (A327S-SU-CF1.00IN-PTF, Olympus) coupled via plastic 20 mL syringe and degassed water as previously described [22]. The plastic portion of the petri dish was removed to couple the transducer directly to the agar slab, and a 2x objective (MRD00025, Nikon) replaced the 4x objective to allow for full imaging of the larger focal area of the transducer.

## Behavioral assays

For experiments with microbubbles, polydisperse microbubbles (PMB, Advanced Microbubbles, Newark, CA) were diluted to a concentration of ~4x10$^7$ and added to an empty 2% agar plate 20 minutes before imaging to allow for absorption/evaporation of the solvent media, leaving microbubbles on the surface of the agar. A dry filter paper with a 1 cm hole previously soaked with 200mM copper sulfate solution was placed around the microbubble lawn, and a young adult *C. elegans* was moved from a home plate to the imaging plate using an eyelash. The agar plate was moved around using the motorized stage to place the worm into the focal zone of the transducer where it was stimulated with a single ultrasound pulse of appropriate amplitude. Videos were recorded for 10 seconds at 10 frames/second, with ultrasound stimulation (described above, via TTL pulse) occurring at 1.5 seconds.

Reversals with more than two head bends were characterized as large reversals, those with fewer than two head bends were characterized as small reversals, and omega bends were those which led to a high-angled turn that lead to a substantial change in direction of movement [21]. Wildtype animals were tested daily to monitor and maintain a baseline level of reversal behavior, and comparisons between strains were made for animals recorded within same days of testing. Where possible, wildtype, mutants and rescue animals were tested on the same day. For 10 MHz ultrasound stimulation, all deviations from the baseline are plotted in S3C Fig. Behavioral data were collected over at least three days to confirm reproducibility, the data were then pooled for final statistical analysis, shown in relevant figures.

## C. elegans

Wildtype *C. elegans*–CGC N2; VC1141 *trp-4(ok1605);* GN716 *trp-4(ok1605)* outcrossed four times [22]; TU253 *mec-4(u253)* [22]; IV903 *trp-4(ok1605); mec-4(u253)*, made by crossing GN716 and TU253.

ASH rescues: IV133 *ueEx71 [Psra-6::trp-4; Pelt-2::GFP]* made by injecting N2 with 50ng/μL *Psra-6::trp-4*, 10ng/μL *elt-2::gfp* for ASH overexpression of *trp-4*. IV160 *trp-4(ok1605) I; ueEx88 [Psra-6::trp4; Pelt-2::GFP]* made by injecting VC1141 with 50ng/μL *Psra-6::trp-4*, 10ng/μL *elt-2::gfp* for ASH rescue of *trp-4*. IV840 *mec-4(u253) X; ueEx71 [Psra-6::trp-4; Pelt-2::GFP]* made by crossing IV133 and TU253.

AWC rescues: IV157 *ueEx85 [Podr-3::trp-4; Pelt-2::GFP]* made by injecting N2 with 50ng/μL *Podr-3::trp4*, 10ng/μL *elt-2::gfp* for AWC overexpression of *trp-4*. IV162 *trp-4(ok1605) I; ueEx89 [Podr-3::trp-4; Pelt-2::GFP]* made by injecting VC1141 with 50ng/μL *Podr-3::trp-4*, 10ng/μL *elt-2::gfp* for AWC rescue of *trp-4*. IV839 *mec-4(u253) X; ueEx85 [Podr-3::trp-4; Pelt-2::GFP]* made by crossing IV157 and TU253.

## Behavioral ultrasound pressure and temperature measurements

Pressure and temperature measurements were collected through 2% agar plates using a Precision Acoustics Fiber-Optic Hydrophone connected to a 1052B Oscilloscope (Tektronix, Beaverton, OR). The hydrophone probe was moved sub-mm distances using the same stage used for animal recordings while the petri dish was held in place using a three-prong clamp. The noise floor for this instrument is 10kPa [44], and the uncertainty of the instrument is 10% in the frequency range used in this study [45], allows us to adequately measure pressures used in this study (>500kPa). The full width at half maximum (FWHM) reported was determined by interpolating the perpendicular distances at which half of the maximum pressure was measured using the fiberoptic hydrophone, and averaging over the amplifier gains used to achieve various pressures.

## Statistical analysis

Data were analyzed by combining several days' recordings within several days' experimental sessions, with sample sizes chosen to reflect those described previously [21]. All behavioral data were plotted as proportion of response plus/minus standard error of the proportion. For significance tests, two-proportion z-tests were used with Bonferroni corrections for multiple comparisons. All sample sizes were >30, and animals were chosen at random from their broader population. The observer was not blind to the genotype of the group being tested. Animals were excluded from the study if they showed visible signs of injury upon transfer to the assay.

## Supporting information

**S1 Fig. Frequency of small reversals, defined as fewer than two head bends, and omega bends in different strains at different peak negative pressures.** (A, C, E) Small reversals and (B, D, F) omega bends from Fig 2 recordings. n = 45 for each condition (n = 90–135 in c-d single mutants). Proportion of animals responding with standard error of the proportion are shown. *** $p < .001$, ** $p < .01$, * $p < .05$ by two-proportion z-test with Bonferroni correction (c = 5) for multiple comparisons.
(TIF)

**S2 Fig. Frequency of small reversals, defined as fewer than two head bends, and omega bends in different strains at different peak negative pressures.** (A, C, E) Small reversals and (B, D, F) omega bends from Fig 3 recordings. n = 45 for each condition, Proportion of animals responding with standard error of the proportion are shown. *** $p < .001$, ** $p < .01$, * $p < .05$ by two-proportion z-test with Bonferroni correction (c = 3) for multiple comparisons.
(TIF)

**S3 Fig. Schematic of 10 MHz behavioral imaging setup, modified from Kubanek et al. 2018 [22].** (A) Experimental setup has agar slab with *C. elegans* corralled by a copper sulfate barrier resting on top of a 20 mL syringe. Degassed water couples the piezoelectric line-focused transducer (10 MHz) to the agar slab. (B) Hydrophone measurements at different perpendicular positions relative to the transducer line focus, with peak negative pressures reaching 1 MPa at highest amplifier settings. Yellow bar represents line focus of highest pressure, points connected via spline fit, FWHM ~ 0.52mm. (C) *C. elegans* exhibits minimal behavioral responses to 10 MHz ultrasound stimuli, although these are significantly enhanced in the presence of microbubbles. (D) Example image of *C. elegans* on agar slab approaching ultrasound focal line. Yellow dots (some highlighted for visibility) represent head positions of each of n = 224 worms, indicating a significant proportion of ultrasound stimulations occurred when the head was positioned within the high-pressure band (yellow).
(TIF)

**S1 Video. WT *C. elegans* performing a large reversal (and omega bend) in response to ultrasound.** 10 second video recorded at 10 fps, with ultrasound pulse at 1.5 s (15 frames). Ultrasound label temporally extended for visibility. After ultrasound pulse, worm reverses with >2 head bends and completes an omega bend reorientation.
(MP4)

**S1 Data. Minimal data set for Magaram et al. 2022.** An excel file with minimal data for Figs 1–3 and S1–S3.
(XLSX)

## Acknowledgments

We thank S. Xu, M. Goodman and the CGC for strains. We also thank J. Kubanek for technical advice and A. Singh, and members of the Chalasani and Friend labs for helpful comments and suggestions on the manuscript.

## Author Contributions

**Conceptualization:** Uri Magaram, Connor Weiss, Sreekanth H. Chalasani.

**Data curation:** Uri Magaram, Connor Weiss.

**Formal analysis:** Uri Magaram, Connor Weiss.

**Investigation:** Uri Magaram, Connor Weiss, Aditya Vasan, Kirthi C. Reddy.

**Methodology:** Uri Magaram, Connor Weiss, Aditya Vasan, Kirthi C. Reddy.

**Resources:** Kirthi C. Reddy, Sreekanth H. Chalasani.

**Software:** Uri Magaram.

**Supervision:** James Friend, Sreekanth H. Chalasani.

**Validation:** Uri Magaram.

**Visualization:** Uri Magaram.

**Writing – original draft:** Uri Magaram, Sreekanth H. Chalasani.

**Writing – review & editing:** Sreekanth H. Chalasani.

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
