## [Decision Letter · Decision Letter 0]

9 Dec 2021

PONE-D-21-33415Two parallel pathways are required for ultrasound-evoked behavioral changes in Caenorhabditis elegansPLOS ONE

Dear Dr. Chalasani,

Thank you for submitting your manuscript to PLOS ONE. After careful consideration, we feel that it has merit but does not fully meet PLOS ONE’s publication criteria as it currently stands. Therefore, we invite you to submit a revised version of the manuscript that addresses the points raised during the review process.

I believe that the comments from both reviewers will considerably improve the manuscript. One thing I noticed is that you did not include trp-4;mec-4 double in Fig 2A, which would make a better comparison with Fig 2B (In the Fig. 2 and 3 graph insets, you should use a dash between mec and 4. There should be no space between gene name and allele name). One question I have is why you chose AWC neurons among other sensory neurons for trp-4 rescue experiments.

We look forward to receiving your revised manuscript.

Kind regards,

Hongkyun Kim

Academic Editor

PLOS ONE

Journal Requirements:

"We thank S. Xu, M. Goodman and the CGC for strains. We also thank J. Kubanek for technical advice and A. Singh, and members of the Chalasani and Friend labs for helpful comments and suggestions on the manuscript. This work was funded by grants from the National Institutes of Health (R01MH111534, R01NS115591) (S.H.C.) and from the W.M. Keck Foundation (J.F.),"

""S.H.C.

R01MH111534, R01NS115591

National Institutes of Health

www.nimh.nih.gov, www.ninds.nih.gov

Reviewers' comments:

Reviewer's Responses to Questions

**Comments to the Author**

1. Is the manuscript technically sound, and do the data support the conclusions?

Reviewer #1: Partly

Reviewer #2: Partly

2. Has the statistical analysis been performed appropriately and rigorously? 

Reviewer #1: I Don't Know

Reviewer #2: No

3. Have the authors made all data underlying the findings in their manuscript fully available?

Reviewer #1: Yes

Reviewer #2: No

4. Is the manuscript presented in an intelligible fashion and written in standard English?

Reviewer #1: Yes

Reviewer #2: Yes

5. Review Comments to the Author

Reviewer #1: Uri and colleagues report in this manuscript a follow-up study on the molecular underpinning of C. elegans response to ultrasounds. The group published that the mechanosensory TRP channel TRP-4 is at least in part responsible for the worm response to ultrasounds. Later though, another group reported that another mechanosensory channel of the DEG/ENaC superfamily called MEC-4 was underlying the animals’ response to ultrasounds. Thus, the question that the authors try to address here is whether these two channels function within the same or two parallel molecular pathways. I believe that the authors have data supporting the conclusion that these two channels function in parallel pathways. However, I think the manuscript could be substantially improved.

1. 0.92 MPa seems to be already at the plateau of the response. It would be useful to include a point between 0.79 and 0.92 MPa. This could help in teasing out differences between the mutants.

2. The trp-4 x4 strain should be used for the experiments and the rescues since it is the strain with the least likelihood of carrying other mutations in the background. It is not clear what strain was used for the rescues for example.

3. Rescue experiments both in mec-4 and trp-4 mutants should be performed using mec-4 DNA.

4. Each figure legend should contain the exact number of animals used in each experiment and how many times the experiment was repeated. Also, the figures should show the individual data points and the averages +/- SE or SD.

5. It is intriguing that the expression of TRP-4 in AWC rescues reversals at 0.79 MPa but expression of the same channel in ASH rescues at 0.92 MPa. The authors should comment on this finding and propose a hypothesis.

6. The authors should carefully review the text and the labels of the figures for the C. elegans genetic nomenclature. There are several errors.

7. Please labels the panels within the figures so that it is clear which experiments were done with and which without microbubbles.

8. Line 131, the authors need to explain why they were “unable to replicate this study”. Was it part of the study? The entire study? This is somewhat concerning.

Reviewer #2: In this study, the authors exposed C. elegans to 10 ms pulses of 2.25 MHz, mid-intensity ultrasound with the aid of gas-filled microbubbles. In some conditions, responses without the bubbles were also tested.

The authors replicated previous findings [22, 38] that mutations of the mec-4 pore-forming subunit diminish the responsiveness to ultrasound. The authors also replicated previous findings [22, 38] that mutations of the trp-4 pore-forming subunit have small but likely significant effect on the responses. For one pressure level, but not others, there was a significant increase in responsiveness when trp-4 was overexpressed in AWC neurons.

This study provides a useful replication of previous studies and thus should be eventually published.

There are, nonetheless, several points that should be addressed:

1) The trp-4 (ok1605 4X) effect is small, like in [22] and [38]. This should be acknowledged. There is a suggestion in this paper that the effect may reach significance. The significance values should be provided in the text. Importantly, there are many comparisons made among the numerous bars in the plots. Such multiple comparisons should be corrected for (e.g., Bonferroni), or an omnibus statistical analysis (n-way ANOVA) performed.

2) Fig. 3A versus B reveal enormous variability (20% versus 70%) in the response of WT animals at 0.79 MPa. This raises questions regarding the reliability of assessing the responses, and so the reliability of the plots. It seems the videos were quantified by eye. This should be specified and the reasons for the variability addressed.

3) The rescue AWC::trp-4 was performed on the trp-4 (ok1605), which has many other mutations present, unless these are outcrossed (Figure 2 and [22, 38] confirm this). There can be a complex interactions between the trp-4 and the other affected genes. Therefore, the AWC::trp-4 overexpression may not be conclusive. This should be acknowledged.

4) It seems only Fig. 2A and Suppl. Fig. 2 show data without microbubbles. The authors should specify which datasets used the bubbles.

Additional, less pressing issues:

A. The mec-4 effect appears weaker compared to [22] and [38]. Possible reasons could be provided.

B. "While we are unable to replicate this study" [22]

There are several important technological differences in the present study:

1) There is a relatively broad focus (Figure S2B), likely activating the entire animal. In this case, there may both reversal and acceleration tendencies, cancelling each other out.

2) The slab/dish sit directly over the syringe top, which may not have allowed the assurance that there is continuous water coupling. Water evaporation degrades the coupling over time.

3) [22] used longer duration (300 ms) and pulsed stimuli (1 kHz), which are both critical.

These differences should be acknowledged.

Moreover, the effects of [22] were replicated in two other studies ([38], [39]). These two studies should be cited.

C. mec-4 and trp-4 likely "act in parallel".

The finding of an increased effect does not necessarily imply parallel (or serial) engagement. I suggest to rephrase as "summed" or "compounded" effect, or "both act".

Minor:

i. "indirectly include astrocyte signals"

- based on my read of the study, the astrocytic stimulation is direct

ii. "We previously showed that the pore-forming subunit of the mechanosensitive ion channel TRP-4 is required to generate behavioral responses to a single 10 ms pulse of ultrasound generated from a 2.25 MHz focused transducer [21]"

-> "We previously showed that the pore-forming subunit of the mechanosensitive ion channel TRP-4 is required to generate behavioral responses mediated by microbbubles activated by a 10 ms pulse of ultrasound generated from a 2.25 MHz focused transducer [21]"

iii. "confirming ultrasound-triggered control (Sonogenetics)"

-> "confirming ultrasound-triggered, microbubble-mediated control"

iv. "We generated a trp-4 mec-4 double mutant and compared its ultrasound responses to both single mutants."

-> missing "We found ..." sentence following this sentence.

v. "lower ultrasound intensities (0.79 MPa)"

-> "pressures"

Refs:

[38] "Ultrasound neuro-modulation chip: activation of sensory neurons in Caenorhabditis elegans by surface acoustic waves"

[39] "Ultrasound activation of mechanosensory ion channels in Caenorhabditis elegans"

6. PLOS authors have the option to publish the peer review history of their article (what does this mean?). If published, this will include your full peer review and any attached files.

Reviewer #1: No

Reviewer #2: No

---

## [Author Response · Author response to Decision Letter 0]

9 Feb 2022

Responses (in black) to the reviewer’s concerns (in blue)

Reviewer #1: Uri and colleagues report in this manuscript a follow-up study on the molecular underpinning of C. elegans response to ultrasounds. The group published that the mechanosensory TRP channel TRP-4 is at least in part responsible for the worm response to ultrasounds. Later though, another group reported that another mechanosensory channel of the DEG/ENaC superfamily called MEC-4 was underlying the animals’ response to ultrasounds. Thus, the question that the authors try to address here is whether these two channels function within the same or two parallel molecular pathways. I believe that the authors have data supporting the conclusion that these two channels function in parallel pathways. However, I think the manuscript could be substantially improved.

We thank the reviewer for their positive feedback.

1. 0.92 MPa seems to be already at the plateau of the response. It would be useful to include a point between 0.79 and 0.92 MPa. This could help in teasing out differences between the mutants.

We have included data from 0.88 MPa in this revised manuscript. 

2. The trp-4 x4 strain should be used for the experiments and the rescues since it is the strain with the least likelihood of carrying other mutations in the background. It is not clear what strain was used for the rescues for example.

We have clarified the use of original versus 4x strain in the manuscript, and have included new data in the trp-4 outcrossed strain in the revised manuscript as suggested by the reviewer.

3. Rescue experiments both in mec-4 and trp-4 mutants should be performed using mec-4 DNA.

We agree with the reviewer that this would be an excellent experiment. However, the site of action for the MEC-4 protein has not been identified. We feel that mapping the site of MEC-4 action is beyond the scope of this study.

4. Each figure legend should contain the exact number of animals used in each experiment and how many times the experiment was repeated. Also, the figures should show the individual data points and the averages +/- SE or SD.

We have included the exact number of animals in each experiment in the revised manuscript. However, the animals either respond or don’t to a given ultrasound stimuli, making it difficult to present individual data points. We suggest that the proportion of responses is a better approach to display these results. Error bars represent standard error of the proportion, indicated in the figure legends.

5. It is intriguing that the expression of TRP-4 in AWC rescues reversals at 0.79 MPa but expression of the same channel in ASH rescues at 0.92 MPa. The authors should comment on this finding and propose a hypothesis.

We agree that this is an surprising finding. We have included a hypothesis in line 133-135. We suggest that AWC and ASH sensory neurons might have different sensitivities to mechanical stimuli. 

6. The authors should carefully review the text and the labels of the figures for the C. elegans genetic nomenclature. There are several errors.

We have fixed the errors in C. elegans nomenclature in the revised text, figures, and legends. 

7. Please labels the panels within the figures so that it is clear which experiments were done with and which without microbubbles.

We have included the labels within the panels of each figure to indicate the presence or absence of microbubbles.

8. Line 131, the authors need to explain why they were “unable to replicate this study”. Was it part of the study? The entire study? This is somewhat concerning.

We are unable to replicate C. elegans behavioral responses to 10 MHz ultrasound. We suspect that this might be a result of differences in assay setups and/or ultrasound delivery. We have included this explanation in our revised manuscript (line 217-220). We are able to replicate other parts of the Kubanek et al manuscript (C. elegans responds to ultrasound without microbubbles, mec-4 and trp-4 mutants are defective in their ultrasound-evoked behaviors).

Reviewer #2: In this study, the authors exposed C. elegans to 10 ms pulses of 2.25 MHz, mid-intensity ultrasound with the aid of gas-filled microbubbles. In some conditions, responses without the bubbles were also tested.

The authors replicated previous findings [22, 38] that mutations of the mec-4 pore-forming subunit diminish the responsiveness to ultrasound. The authors also replicated previous findings [22, 38] that mutations of the trp-4 pore-forming subunit have small but likely significant effect on the responses. For one pressure level, but not others, there was a significant increase in responsiveness when trp-4 was overexpressed in AWC neurons.

This study provides a useful replication of previous studies and thus should be eventually published.

We thank the reviewer for their positive feedback.

There are, nonetheless, several points that should be addressed:

1) The trp-4 (ok1605 4X) effect is small, like in [22] and [38]. This should be acknowledged. There is a suggestion in this paper that the effect may reach significance. The significance values should be provided in the text. Importantly, there are many comparisons made among the numerous bars in the plots. Such multiple comparisons should be corrected for (e.g., Bonferroni), or an omnibus statistical analysis (n-way ANOVA) performed.

Our study shows that the trp-4(ok1605 4x) has significant behavioral defects to specific ultrasound pressures. We have added a description of these data in the revised manuscript (line 98). Also, we add appropriate statistical measures corrected for multiple comparisons, described in all relevant figure legends. 

2) Fig. 3A versus B reveal enormous variability (20% versus 70%) in the response of WT animals at 0.79 MPa. This raises questions regarding the reliability of assessing the responses, and so the reliability of the plots. It seems the videos were quantified by eye. This should be specified and the reasons for the variability addressed.

We thank the reviewer for catching this discrepancy. Indeed, this was a mistaken data point (small reversals were plotted instead of large reversals) and the panel has been updated with the correct data that shows consistent N2 behavior across the AWC and ASH rescue experiments.

3) The rescue AWC::trp-4 was performed on the trp-4 (ok1605), which has many other mutations present, unless these are outcrossed (Figure 2 and [22, 38] confirm this). There can be a complex interactions between the trp-4 and the other affected genes. Therefore, the AWC::trp-4 overexpression may not be conclusive. This should be acknowledged.

We include rescue experiments in the trp-4(ok1605 4x) outcrossed strain in the revised manuscript (new data). We find that the expressing trp-4 in AWC or ASH does indeed rescue the behavioral deficits (at 0.99 MPa ultrasound) of the outcrossed strain. 

4) It seems only Fig. 2A and Suppl. Fig. 2 show data without microbubbles. The authors should specify which datasets used the bubbles.

We have included labels within the panels of each figure to indicate the presence or absence of microbubbles.

Additional, less pressing issues:

A. The mec-4 effect appears weaker compared to [22] and [38]. Possible reasons could be provided.

Our experiments use ultrasound at 2.25 MHz, while [22] uses 10 MHz ultrasound. [38] makes no reference to mec-4, but uses 28 MHz ultrasound. Also, [39] tests a different allele of trp-4. It is possible that there are different mechanisms at play at different ultrasound frequencies. We have included this explanation in our revised manuscript (line 220-226). 

B. "While we are unable to replicate this study" [22]

There are several important technological differences in the present study:

1) There is a relatively broad focus (Figure S2B), likely activating the entire animal. In this case, there may both reversal and acceleration tendencies, cancelling each other out.

2) The slab/dish sit directly over the syringe top, which may not have allowed the assurance that there is continuous water coupling. Water evaporation degrades the coupling over time.

3) [22] used longer duration (300 ms) and pulsed stimuli (1 kHz), which are both critical.

These differences should be acknowledged.

Moreover, the effects of [22] were replicated in two other studies ([38], [39]). These two studies should be cited.

We agree that there are differences in ultrasound delivery between our present study and these previous studies. We have included these explanations in our revised manuscript and cited these two studies (line 220-224). 

C. mec-4 and trp-4 likely "act in parallel".

The finding of an increased effect does not necessarily imply parallel (or serial) engagement. I suggest to rephrase as "summed" or "compounded" effect, or "both act".

We have rephrased “act in parallel” to “summed” or “both act”.

Minor:

i. "indirectly include astrocyte signals"

- based on my read of the study, the astrocytic stimulation is direct

This study says that the effect on the neurons is indirect as it is a result of a change in astrocyte signals. Edited this line.

ii. "We previously showed that the pore-forming subunit of the mechanosensitive ion channel TRP-4 is required to generate behavioral responses to a single 10 ms pulse of ultrasound generated from a 2.25 MHz focused transducer [21]"

-> "We previously showed that the pore-forming subunit of the mechanosensitive ion channel TRP-4 is required to generate behavioral responses mediated by microbbubles activated by a 10 ms pulse of ultrasound generated from a 2.25 MHz focused transducer [21]"

Fixed.

iii. "confirming ultrasound-triggered control (Sonogenetics)"

-> "confirming ultrasound-triggered, microbubble-mediated control"

Fixed.

iv. "We generated a trp-4 mec-4 double mutant and compared its ultrasound responses to both single mutants."

-> missing "We found ..." sentence following this sentence.

Fixed.

v. "lower ultrasound intensities (0.79 MPa)"

-> "pressures"

Fixed.

Refs:

[38] "Ultrasound neuro-modulation chip: activation of sensory neurons in Caenorhabditis elegans by surface acoustic waves"

[39] "Ultrasound activation of mechanosensory ion channels in Caenorhabditis elegans"

---

## [Decision Letter · Decision Letter 1]

4 Mar 2022

PONE-D-21-33415R1Two pathways are required for ultrasound-evoked behavioral changes in Caenorhabditis elegansPLOS ONE

Dear Dr. Chalasani, Thank you for submitting your manuscript to PLOS ONE. After careful consideration, we feel that it has merit but does not fully meet PLOS ONE’s publication criteria as it currently stands. Therefore, we invite you to submit a revised version of the manuscript that addresses the points raised during the review process. Both reviewers requested minor revisions, which can be readily addressed.

We look forward to receiving your revised manuscript.

Kind regards,

Hongkyun Kim

Academic Editor

PLOS ONE

Journal Requirements:

Reviewers' comments:

Reviewer's Responses to Questions

**Comments to the Author**

1. If the authors have adequately addressed your comments raised in a previous round of review and you feel that this manuscript is now acceptable for publication, you may indicate that here to bypass the “Comments to the Author” section, enter your conflict of interest statement in the “Confidential to Editor” section, and submit your "Accept" recommendation.

Reviewer #1: All comments have been addressed

Reviewer #2: (No Response)

2. Is the manuscript technically sound, and do the data support the conclusions?

Reviewer #1: Yes

Reviewer #2: Yes

3. Has the statistical analysis been performed appropriately and rigorously? 

Reviewer #1: Yes

Reviewer #2: I Don't Know

4. Have the authors made all data underlying the findings in their manuscript fully available?

Reviewer #1: No

Reviewer #2: No

5. Is the manuscript presented in an intelligible fashion and written in standard English?

Reviewer #1: Yes

Reviewer #2: Yes

6. Review Comments to the Author

Reviewer #1: This is a revised manuscript from Maragam and colleagues, in which they explored the idea that mechanosensitive channels mec-4 and trp-4 might function together in detecting ultrasounds. The authors have addressed the concerns we raised in the first round of reviews. There are a few minor points that still need to be addressed.

1. In the abstract, results and discussion, the authors removed the word “parallel” and replaced it with “together”. However, “parallel pathways” is still listed under keywords. It should be removed.

2. Figure 2B, 0.88 MPa group. Is there statistical difference between the strains? if so, statistics need to be added.

3. Trp-4 over-expression in AWC neurons of mec-4 mutants rescues the phenotype. These two channels belong to two different families. Based on what it is known about their mechanosensitivity and physiological properties, can the authors add some discussion on how they envision this rescue being possible?

4. There should be a space between number and unit. Check for the presence of the space throughout the manuscript. For example the space is missing at these two locations: Line 145: 28MHz, line 169: 2.25MHz.

Reviewer #2: The authors addressed most of the comments, and this will be an interesting study.

There are two final points to be addressed:

A) The Bonferroni correction should say how many comparisons c were considered, e.g., by what value was the p-value divided by to yield the ultimate p/c value, stars for which are shown in the figures.

B) A set of points associated with the 10 MHz data (Suppl Fig. 3).

Given the difficulties in obtaining robust responses, it is important to stress the differences from [28] so that others can replicate these findings:

i) SFig. 3A shows a white plastic piece below the agar slab. In [28], no such piece was present---the slab was sitting directly over the "syringe top". In the present study, the presence of this piece may complicate coupling (air pockets forming within it) and its monitoring.

ii) Sfig. 3B shows that the setup produced a relatively [28] large focus. The full-width half-maximum (FWHM) values should be described. It appears that the average FWHM was on the order of the animal's length. This way, most of the animal was stimulated, precluding effective frontal stimulation that leads to reliable reversals.

iii) Sfig. 3C would be stronger if it included all responses; not just large reversals; ([28] used all responses).

iv) In [28], responses were quantified objectively using an algorithm run on recorded videos. Thus, experimenters were blinded to the results. In the present study, the calls were made by the naked eye.

v) Besides the subjective judgements in the present study, the eye may not be sensitive enough to spot effects. The algorithm in [28] assessed *any* deviation from baseline; not just reversals.

These points should be stressed in the Methods and the Results.

7. PLOS authors have the option to publish the peer review history of their article (what does this mean?). If published, this will include your full peer review and any attached files.

Reviewer #1: No

Reviewer #2: No

---

## [Author Response · Author response to Decision Letter 1]

19 Mar 2022

Responses (in black) to the reviewers comments (in blue)

Reviewer #1: This is a revised manuscript from Maragam and colleagues, in which they explored the idea that mechanosensitive channels mec-4 and trp-4 might function together in detecting ultrasounds. The authors have addressed the concerns we raised in the first round of reviews. There are a few minor points that still need to be addressed.

1. In the abstract, results and discussion, the authors removed the word “parallel” and replaced it with “together”. However, “parallel pathways” is still listed under keywords. It should be removed.

Fixed.

2. Figure 2B, 0.88 MPa group. Is there statistical difference between the strains? if so, statistics need to be added.

There is no statistical differences between the strains in the 0.88 MPa group. We have edited the figure appropriately.

3. Trp-4 over-expression in AWC neurons of mec-4 mutants rescues the phenotype. These two channels belong to two different families. Based on what it is known about their mechanosensitivity and physiological properties, can the authors add some discussion on how they envision this rescue being possible?

Included a comment. Line 130-134

4. There should be a space between number and unit. Check for the presence of the space throughout the manuscript. For example the space is missing at these two locations: Line 145: 28MHz, line 169: 2.25MHz.

Fixed.

Reviewer #2: The authors addressed most of the comments, and this will be an interesting study.

There are two final points to be addressed:

A) The Bonferroni correction should say how many comparisons c were considered, e.g., by what value was the p-value divided by to yield the ultimate p/c value, stars for which are shown in the figures.

We have added the Bonferroni correction along with the number of comparisons (c) in the appropriate figures. We have made changes to other figures to show the corrected p-values.

B) A set of points associated with the 10 MHz data (Suppl Fig. 3).

Given the difficulties in obtaining robust responses, it is important to stress the differences from [28] so that others can replicate these findings:

i) SFig. 3A shows a white plastic piece below the agar slab. In [28], no such piece was present---the slab was sitting directly over the "syringe top". In the present study, the presence of this piece may complicate coupling (air pockets forming within it) and its monitoring.

ii) Sfig. 3B shows that the setup produced a relatively [28] large focus. The full-width half-maximum (FWHM) values should be described. It appears that the average FWHM was on the order of the animal's length. This way, most of the animal was stimulated, precluding effective frontal stimulation that leads to reliable reversals.

iii) Sfig. 3C would be stronger if it included all responses; not just large reversals; ([28] used all responses).

iv) In [28], responses were quantified objectively using an algorithm run on recorded videos. Thus, experimenters were blinded to the results. In the present study, the calls were made by the naked eye.

v) Besides the subjective judgements in the present study, the eye may not be sensitive enough to spot effects. The algorithm in [28] assessed *any* deviation from baseline; not just reversals.

These points should be stressed in the Methods and the Results.

We do not have a plastic piece between the agar surface and the 10 MHz transducer (clarified in the methods). The image in Supplementary Figure 3A is showing the top of the plastic syringe. We have now marked this in the figure for clarity. We only scored animals where we stimulated the head. In Supplementary figure 3B, the area shaded is about 0.4 mm at the maximum, which is half the length of the animal. Further, in Supplementary Figure S3D, we are showing all the positions of the head of the animal in our assay. These data indicate a broad distribution, but we were unable to observe any significant behavioral effects. Additionally, the Kubanek et al 2018 study used a hydrophone to confirm that the agar did not attenuate the ultrasound pressure, the shape of the radiation force was plotted used a simulation (which is likely to be imprecise), while we are reporting measurements from a Precision Acoustics fiber optic probe across the agar surface. 

We now plot all responses and not just large reversals in Supplementary Figure S3C. While we did score these behaviors by eye, we did not observe any significant deviation from the baseline. Furthermore, in Figure 2B from Kubanek et al 2018, the reversals observed appear to last multiple seconds, which we would have definitely identified in our new analysis. 

However, we do include a comment in the methods section. Line 215.

---

## [Decision Letter · Decision Letter 2]

4 Apr 2022

PONE-D-21-33415R2Two pathways are required for ultrasound-evoked behavioral changes in Caenorhabditis elegansPLOS ONE

Dear Dr. Chalasani,

Thank you for submitting your manuscript to PLOS ONE. After careful consideration, we feel that it has merit but does not fully meet PLOS ONE’s publication criteria as it currently stands. Therefore, we invite you to submit a revised version of the manuscript that addresses the points raised during the review process.

I believe that you can readily address reviewer #2's comments on the quantification of Supplementary Fig. 3B.

Please submit your revised manuscript by April 15th or earlier. I will expedite the process as soon as possible. If you will need more time than this to complete your revisions, please reply to this message or contact the journal office at plosone@plos.org. Please include the following items when submitting your revised manuscript:A rebuttal letter that responds to each point raised by the academic editor and reviewer(s). You should upload this letter as a separate file labeled 'Response to Reviewers'.A marked-up copy of your manuscript that highlights changes made to the original version. You should upload this as a separate file labeled 'Revised Manuscript with Track Changes'.An unmarked version of your revised paper without tracked changes. You should upload this as a separate file labeled 'Manuscript'.If applicable, we recommend that you deposit your laboratory protocols in protocols.io to enhance the reproducibility of your results. Protocols.io assigns your protocol its own identifier (DOI) so that it can be cited independently in the future. For instructions see: https://journals.plos.org/plosone/s/submission-guidelines#loc-laboratory-protocols. Additionally, PLOS ONE offers an option for publishing peer-reviewed Lab Protocol articles, which describe protocols hosted on protocols.io. Read more information on sharing protocols at https://plos.org/protocols?utm_medium=editorial-email&utm_source=authorletters&utm_campaign=protocols.

We look forward to receiving your revised manuscript.

Kind regards,

Hongkyun Kim

Academic Editor

PLOS ONE

Journal Requirements:

Reviewers' comments:

Reviewer's Responses to Questions

**Comments to the Author**

1. If the authors have adequately addressed your comments raised in a previous round of review and you feel that this manuscript is now acceptable for publication, you may indicate that here to bypass the “Comments to the Author” section, enter your conflict of interest statement in the “Confidential to Editor” section, and submit your "Accept" recommendation.

Reviewer #1: All comments have been addressed

Reviewer #2: (No Response)

2. Is the manuscript technically sound, and do the data support the conclusions?

Reviewer #1: Yes

Reviewer #2: Yes

3. Has the statistical analysis been performed appropriately and rigorously? 

Reviewer #1: Yes

Reviewer #2: Yes

4. Have the authors made all data underlying the findings in their manuscript fully available?

Reviewer #1: Yes

Reviewer #2: No

5. Is the manuscript presented in an intelligible fashion and written in standard English?

Reviewer #1: Yes

Reviewer #2: Yes

6. Review Comments to the Author

Reviewer #1: (No Response)

Reviewer #2: I appreciate the authors' response. There are now only minor points to be addressed.

1. The PA fiber-optic hydrophone is inaccurate for measurements of the relatively small pressured used in this study. Estimates of its accuracy for these pressures should be provided. A dB value suffices.

2. The 10 MHz setup differs from that of Kubanek et al. 2018. In the present study, the agar sits directly on a syringe. This way, there is no way to validate good and continued coupling.

At the very least, the caption of SF3 should be changed from

"Schematic of 10 MHz behavioral imaging setup as described in Kubanek et al 2018."

to

Schematic of 10 MHz behavioral imaging setup."

3. My previous suggestion to quantify the FWHM in SF3B does not seem to be taken. Yet, the focality of the field is critical for effective C. elegans responses and thus should be quantified.

4. In SF3C, it is worth to indicate effect significance (e.g., using stars).

7. PLOS authors have the option to publish the peer review history of their article (what does this mean?). If published, this will include your full peer review and any attached files.

Reviewer #1: No

Reviewer #2: No

---

## [Author Response · Author response to Decision Letter 2]

12 Apr 2022

Dear Dr. Kim

Please find attached a reviewer response file along with the manuscript file with and without tracked changes. We would like you to consider the comments from reviewer 2 in totality with this being our third revision. They seem to be resistant to our manuscript and are bringing up newer concerns in each subsequent review (for example, they are now asking for an estimate of the accuracy of our hydrophone, edit our figure title, etc.). 

We would request you to make a decision on whether this reviewer is being reasonable. Also, we would like an opportunity to discuss our concerns with you.

Sincerely

Sreekanth Chalasani and Uri Magaram

Salk Institute

La Jolla, CA

Responses in black to the reviewers comments in blue

1. The PA fiber-optic hydrophone is inaccurate for measurements of the relatively small pressured used in this study. Estimates of its accuracy for these pressures should be provided. A dB value suffices.

We have updated the manuscript based on the reviewers recommendation and confirming that the manufacturer has guaranteed operation in frequency range used in this study. Further, additional peer-reviewed work by several groups have used this system in the desired frequency range with similar levels of acoustic pressure [1,2].

There is no direct conversion to dB as the uncertainty depends on the original measurement amplitude. However, for a 1 MPa signal, the uncertainty would be 100 kPa. We have included the relevant citations in the methods section which justify the use of the PA fiber-optic hydrophone. 

2. The 10 MHz setup differs from that of Kubanek et al. 2018. In the present study, the agar sits directly on a syringe. This way, there is no way to validate good and continued coupling.

At the very least, the caption of SF3 should be changed from

"Schematic of 10 MHz behavioral imaging setup as described in Kubanek et al 2018."

to

Schematic of 10 MHz behavioral imaging setup."

Edited the caption 

3. My previous suggestion to quantify the FWHM in SF3B does not seem to be taken. Yet, the focality of the field is critical for effective C. elegans responses and thus should be quantified.

As recommended by the reviewer, the FWHM in SF3B has been added to the figure legend and relevant methods section. The FWHM we observed is on the order of half the body length of the animal, which we believe gives us enough precision to be able to stimulate the head without the tail as we know is necessary from our own experience and previous literature. Additionally, for this reason we went back and marked the head positions of all the worms in the assay at the moment of stimulation, displayed in SF3D to ensure that the ultrasound stimulus was delivered to the head of the animal as best as possible.

4. In SF3C, it is worth to indicate effect significance (e.g., using stars).

Added as recommended. We find that behavioral responses in the presence of microbubbles is significantly more than those in their absence and also referenced this in the manuscript lines 148-151. 

References

1. Lakshmanan A, Jin Z, Nety SP, Sawyer DP, Lee-Gosselin A, Malounda D, Swift MB, Maresca D, Shapiro MG. Acoustic biosensors for ultrasound imaging of enzyme activity. Nat Chem Biol. 2020 Sep;16(9):988-996. doi: 10.1038/s41589-020-0591-0. Epub 2020 Jul 13. Erratum in: Nat Chem Biol. 2020 Jul 23;: PMID: 32661379; PMCID: PMC7713704.

2. Vasan, A., Allein, F., Duque, M., Magaram, U., Boechler, N., Chalasani, S.H. and Friend, J. (2022), Microscale Concert Hall Acoustics to Produce Uniform Ultrasound Stimulation for Targeted Sonogenetics in hsTRPA1-Transfected Cells. Adv. NanoBiomed Res. 2100135. https://doi.org/10.1002/anbr.202100135

---

## [Decision Letter · Decision Letter 3]

14 Apr 2022

Two pathways are required for ultrasound-evoked behavioral changes in Caenorhabditis elegans

PONE-D-21-33415R3

Dear Dr. Dr Chalasani,

We’re pleased to inform you that your manuscript has been judged scientifically suitable for publication and will be formally accepted for publication once it meets all outstanding technical requirements.

Kind regards,

Hongkyun Kim

Academic Editor

PLOS ONE

Additional Editor Comments (optional):

Reviewers' comments:

Reviewer's Responses to Questions

**Comments to the Author**

1. If the authors have adequately addressed your comments raised in a previous round of review and you feel that this manuscript is now acceptable for publication, you may indicate that here to bypass the “Comments to the Author” section, enter your conflict of interest statement in the “Confidential to Editor” section, and submit your "Accept" recommendation.

Reviewer #2: All comments have been addressed

2. Is the manuscript technically sound, and do the data support the conclusions?

Reviewer #2: Yes

3. Has the statistical analysis been performed appropriately and rigorously? 

Reviewer #2: Yes

4. Have the authors made all data underlying the findings in their manuscript fully available?

Reviewer #2: No

5. Is the manuscript presented in an intelligible fashion and written in standard English?

Reviewer #2: Yes

6. Review Comments to the Author

Reviewer #2: I appreciate the authors' response, and this interesting article can now be published.

A final suggestion for an improvement:

"C. elegans exhibits minimal behavioral responses to 10 MHz ultrasound stimuli"

In SF3, I meant to indicate the significance of both bars (instead of a paired test). It would be good to know if the "minimal behavioral response" was significant.

7. PLOS authors have the option to publish the peer review history of their article (what does this mean?). If published, this will include your full peer review and any attached files.

Reviewer #2: No

---

## [Editor Report · Acceptance letter]

27 Apr 2022

PONE-D-21-33415R3 

Two pathways are required for ultrasound-evoked behavioral changes in *Caenorhabditis elegans*

Dear Dr. Chalasani:

I'm pleased to inform you that your manuscript has been deemed suitable for publication in PLOS ONE. Congratulations! Your manuscript is now with our production department. 

Kind regards, 

on behalf of

Dr. Hongkyun Kim 

Academic Editor

PLOS ONE